# Atmospheric electricity observations at Lerwick Geophysical Observatory

R. Giles Harrison[a] and John C. Riddick[b]

[a]Department of Meteorology, Earley Gate, University of Reading, Reading. RG6 6ET, UK

[b]Lockerbie, Scotland. (Formerly with British Geological Survey)

Correspondence to: Giles Harrison (r.g.harrison@reading.ac.uk)

**Abstract**

Atmospheric electricity measurements were made at Lerwick Observatory, Shetland, between 1925 and 1984. These principally provide a long series of hourly Potential Gradient (PG) measurements at an unpolluted site, but also include air-earth current density measurements during the late 1970s and early 1980s. An especially notable aspect was investigating the dramatic atmospheric electrical changes caused by nuclear weapon detonations in the late 1950s and early 1960s, which has parallels with the discovery of the Antarctic ozone hole. The methodology employed at Lerwick to provide the PG measurements is described. There is renewed international interest in such measurements, not least because the Lerwick PG data have been shown to be linked to Pacific Ocean temperature anomalies. The past measurements described have characterised the Lerwick site exceptionally well in atmospheric electrical terms, which also indicate its suitability for future similar measurements.

Keywords: Potential Gradient, conduction current; global circuit; electrograph; ENSO;

## 1. Introduction

Geophysical studies in the UK have a long history, and indeed present one of the longest formal experimental investigations in the natural sciences, arguably beginning with the work of William Gilbert in Tudor times (Gilbert, 1600). Regular and systematic measurements have since provided a range of important insights, many associated with Greenwich and Kew Observatories in London during the nineteenth century. Atmospheric electrical measurements were often pursued alongside magnetic measurements. The expectation of such an arrangement led to atmospheric electricity investigations being included on the geomagnetic survey ship *Carnegie*. The significant result in the *Carnegie* observations was finding a consistent daily variation linked to Universal Time, widely known as the Carnegie curve (Harrison, 2013).

Atmospheric electricity measurements formed part of routine work at observatories across the world during the twentieth century, such as those operated by the Carnegie Institution in South America and western Australia, as

well as by meteorological services in Europe. As further UK magnetic observatories became sought and built beyond London, new sites for regular atmospheric electricity measurements consequently also became established, notably at Eskdalemuir in Scotland and Lerwick in Shetland (Harrison, 2003). Lerwick Observatory has recently celebrated the centenary of its foundation.

Following the renewed international interest in atmospheric electricity measurements for climate research (Nicoll et al., 2019), specific attention is given here to the Lerwick atmospheric electricity instrumentation, which was operational from 1925 to 1984. The Observatory is located just off the A970 main road between Lerwick and Scalloway, on rising land at 80 m above sea level and about 2 km from the town of Lerwick. Figure 1a and 1b shows the location of Lerwick, and Figure 1c and 1d the region around the Observatory including Trebister Loch. Descriptions of the other meteorological work of the site at Lerwick are available in Harper (1950) and Tyldesley (1971). The annual volumes of the *Observatories' Year Book* (hereafter "OYB"), published until 1967, provide a further important resource.

## 2. Development of Lerwick Observatory

### 2.1 Foundation

Expansion of electrical distribution networks towards the end of the nineteenth century generated interference with the work of geomagnetic observatories, requiring more remote sites beyond London to be sought (Macdonald, 2018). These endeavours led to a new observatory being built at Eskdalemuir, Scotland, in 1909, at which the daily recording routine also included atmospheric electricity measurements (Harrison, 2004). With Eskdalemuir Observatory established, the quest for a more northerly site became urgent following the First World War, after a request from the Norwegian government for help in undertaking research into meteorological and auroral phenomena in high latitudes[1] (Walker, 2011a). A permanent site was the preferred option, for which Shetland was especially suitable, as not only was it as far north as possible in the UK, but it also allowed measurements from Kew, Eskdalemuir and Shetland to lie roughly on a north-south line for research on the position and pattern of the electric current systems associated with magnetic disturbances.

Establishing such a northern observatory was included in the programme of the Meteorological Office in 1919. A radio station had been built near Lerwick in 1913 by the Admiralty, and transferred after the war to the Post Office, but was little used. As it offered office space and living quarters, it was well suited to becoming the intended observatory. An arrangement followed with the Air Ministry. It was agreed that the Lerwick radio station site could be used as a geophysical observatory, in return for maintaining the radio equipment, which itself was used for the transmission of meteorological reports and time signals[2]. Lerwick Observatory was formally opened on the 7th June, 1921 by Dr Crichton Mitchell, Superintendent of the Meteorological Office at Edinburgh, with Mr Jock Crichton becoming the first officer in charge.

---

[1] One specific need was to provide high latitude comparative data for Roald Amundsen's arctic voyages. In later years, Lerwick was also used for staff training by the British Antarctic Survey.
[2] The radio heritage is embedded in the site being known locally as "Da Wireless".

## 2.2 Infrastructure developments

At the Observatory's outset, there were thirteen small houses on the site. Only some of these were used for accommodation, allocated to the Superintendent Mr Crichton, the caretaker Mr Ridland, and a wireless operator, Mr Newcomb. Comfort was basic, with official furniture provided from 1926. One house had its scullery used as a darkroom for auroral photographic work, another provided the Observatory office and another a workshop. A further house had darkrooms for use with the electrograph and the photographic recorder of a geomagnetic induction loop around Trebister.

The early electricity supply at the site was poor, and greatly improved by installation of an oil-powered generator in November 1925. Water for the Observatory was obtained from Trebister by a windmill pump, with the water filtered through shingle and sand. The water supply was also initially unreliable as the first pump was inadequately engineered for the strong winds encountered, and an improved wind pump was installed in March 1925.

Early use of the Observatory was strongly influenced by its military past. In 1925 barbed wire entanglements around the site were removed and new paths constructed across the site, from material obtained by cutting drains to prevent flooding. The wireless station was operated for the Air Ministry and used for emergency communications by the General Post Office (GPO) in March 1925, following a cable breakage with the mainland. This occurred again in March 1928. The antenna masts were finally dismantled in September 1932.

With the Second World War looming, a temporary radio station was re-established in July 1939, intended for receiving time signals and weather reports. After the outbreak of war, black-out curtains were fitted to Observatory windows: bombers were seen to pass low over the magnetic huts in an air raid on Lerwick on 22nd November[3]. Some requisition of facilities was made by the army in 1940, which led to difficulties with the water supply, not least from the use of Trebister Loch for washing. The workshop in one of the houses was made into an air-raid shelter in 1941.

New offices were opened in 1960, and a substantial refurbishment led to the opening of a new building in 2013. Office blocks and residential buildings were near to the road, with a meteorological balloon shed on the north-easterly side of the site, overlooking the town of Lerwick (see Figure 2). The meteorological enclosure was behind the office blocks, with various huts housing magnetometers situated further away, along an access path.

## 3. Atmospheric Electricity instruments

By about 1920, operating principles for standardised atmospheric electricity measurements were well established. These measurements were principally to obtain the atmosphere's vertical electric field (generally known in this context as the vertical potential gradient[4], PG), but also included the electrical conductivity of air

---

[3] Sullom Voe, Shetland, was the first part of the UK to be bombed in the Second World War, on 13th November 1939.(Bennett, 2019)

[4] The vertical component of the atmospheric electric field $E_z$ and the potential gradient $F$ are related by $E_z = -F$.

and the vertical conduction current density. The PG was determined from the electric potential measured at a fixed point above the surface, using a sensing electrode of some kind and an electrostatic voltmeter. The probe providing the local air potential did so by actively exchanging charge with its surroundings, and hence was known as an "equaliser" or "collector". Nineteenth century measurements typically used a flame probe equaliser, connected to a mechanical electrometer, for example on a moveable mast at Greenwich Observatory (Airy, 1847), and sited above the cupola at Kew Observatory (Ronalds, 1847). A flame probe sensor was, however, only suitable for intermittent use. This difficulty was removed with the invention of the Kelvin water dropper equaliser in 1859, which equalised the potential with the air through using a fine mist of water supplied from a header tank. By keeping the tank filled with water, continuous measurements of PG became possible, and permanent recordings of the variations obtained were made by projecting the electrometer's deflection onto photographic recording paper (Aplin & Harrison, 2013). Following the naming convention for other meteorological self-registering (or "autographic") instruments[5], a continuous recording device for the PG was known as an "electrograph", or, more specifically when a water dropper equaliser was used, the "Kelvin electrograph". The paper chart records were known as *electrograms*.

For the new site at Lerwick, the intention was also to install an electrograph and Figure 3 shows the approximate positions of the various atmospheric electricity measurement sites. From the outset, a radioactive equaliser was considered more suitable rather than a Kelvin water dropper equaliser. The Lerwick electrograph consisted of a radioactive probe exposed to the atmosphere, connected to an electrometer and chart recorder, initially a Benndorf instrument (Benndorf, 1906), shown in Figure 4a. A design feature of the Benndorf electrometer was the mechanical linkage (Figure 4b), in its original form using a pot of sulphuric acid with a mica vane for damping of oscillations (Hatakeyama, 1934). Figure 4c shows a single day's chart trace from the Lerwick Benndorf electrometer, which includes an earthing test.

The electrograph systems operated in very consistent manner between 1927 and 1985, with the electrograph exposure calibrated to open site exposure by comparison with a stretched wire apparatus (Figure 5a) and electrometer (Figure 5b). Evidence of the original support posts used remains (Figure 5c), hence an almost identical arrangement in the same position could be employed again. In the following sections, further information on the operation of the instrumentation is summarised from different intervals during the measurements. Specific events in the history of these measurements are summarised in Table *1*, which illustrate the considerable care and attention to detail needed to maintain the quality of the measurements. The aspects described below draw on the relevant volumes of the OYB, and unpublished notes of R.A. Hamilton[6] (Hamilton, n.d.)

---

In fair weather, *F* is considered positive, and is typically 100 to 150 Vm$^{-1}$. During precipitation, the PG usually becomes large and variable, and generally increases positively during fog.

[5] e.g, barograph, thermograph, anemograph, hyetograph…

[6] Richard Alexander "Hammy" Hamilton FRSE (1912-1991), (McIntosh, 1991). Lerwick superintendent 1960-1966 and inspirational experimental scientist (Walker, 2011b), strongly influenced by Prof G.M. Dobson when a student in physics at Oxford (Ratcliffe, 1992).

### 3.1 1921-1929

In 1922 the first site for electrograph was established towards the edge of the site, with building of a small wooden hut occupying about 2 m$^2$, with a ridged roof. It contained three brick pillars, one for the recording electrometer, another for the clockwork mechanism, and a third for the absolute electrometer for calibrations. The radioactive collector was a spiral of copper wire coated with radium within an adhesive varnish[7]. This sensor was exposed to the atmosphere at about 1 m from the hut's corner. Great difficulties resulted from damp around the apparatus, and, initially, from the lack of an absolute instrument for comparisons and calibration. An oil stove was kept burning continuously to alleviate the damp, together with small electric bulbs near the supports of the collector rod, which used sulphur insulators.

The Benndorf electrograph[8] arrived from Kew on August 8[th] 1924, and a second electrostatic voltmeter to provide absolute reference observations was provided in the following year, on 30[th] March 1925. The electrograph system consisted of an exposed collector rod connected to the Benndorf electrometer, using fine wire. Satisfactory measurements could not be obtained, due both from the oil stove's fumes, and the hut being inconveniently small and remote. The system was moved on 6th July 1926 to a more accessible position in the office block. There, the collector rod passed through a window in the north-west wall, 1.9 m from the building's corner. The copper spiral collector was 4.76 m above the ground and projected 1.23 m. Timing for the electrograph was taken from the Observatory's standard clock, itself synchronised by a daily time signal.

Initially, this new location for the apparatus was expected to be unsatisfactory as it was likely to encounter distortion of the electric field by the buildings, and it was a large distance (236 m), from the previous site of the hut, where the absolute reference measurements were still made. These concerns turned out to be much less important than first anticipated. Results from the 1925 and 1926 measurements during this transition were not, however, published in full, with only a short summary of the results given in the OYB for the monthly mean PG at 03, 09, 15 and 21 GMT.

The absolute measurements of PG were made by the "stretched wire" method[9], from 1926, from which an reduction factor for the Benndorf electrograph was computed to give the equivalent potential over open ground. The stretched wire was a horizontal length of uninsulated wire hung between insulators on two posts, with the wire many times longer than the height of the posts. This allowed the potential at the wire's height to be largely unaffected by the distorting effects of the supports on the atmospheric electric field. At Lerwick, two stout wooden posts 211 cm in height and 9.48 m apart were used to support the stretched wire, with a collector[10] in the centre, exactly 1 m above the ground. A standardising electrometer (of the Wulf design) was connected to

---

[7] In detail, the collector's copper spiral collector was soldered into the small end of a tapered copper alloy (German Silver) tube, 76 cm long, of triangular cross section. This was attached to an aluminium (Duralumin) tube, 89 cm long and 1.3 cm diameter. The tube entered through a hole, 3.8 cm diameter, into a wooden box (dimensions 38 x 25 x 10 cm), held horizontally between the ends of two metal rods supported by sulphur insulators.

[8] Benndorf electrometer No. 108, manufactured by L. Castagna, Vienna.

[9] The "stretched wire" system has also been known as a "passive wire antenna".

[10] At Lerwick this was initially a burning fuse, and later a radioactive (americium) source.

one end of the wire to determine the potential. Ten to twenty readings were obtained from the electrometer at minute intervals, and the reduction factor derived from comparing the mean of these values with the corresponding mean potential at the collector simultaneously recorded by the Benndorf. Smoothed monthly means of the reduction factors were derived, which were applied to the electrograph measurements.

Ten reduction factor determinations typically were made each month, with, in 1927, values ranging from 1.31 to 1.43. The leakage rate of the system was, as a fraction of the rate of charging, 1/20 in the winter and 1/50 in the summer; this variation was included in the experiments determining the reduction factor, which was therefore ~3% lower in summer than winter. In 1928 the least mean monthly reduction factor was 1.19 and the greatest was 1.41. An attempt was made to relate the reduction factor to the wind direction, as larger values were associated with winds from the NE, S, SW, and W, in which directions the electrograph's collector had a good exposure. In other directions flow was obstructed by buildings, and the extent of the reduction in the factor depended on the nearness of the obstructions to the collector. However, the effect was small, with the lowest mean values 1.24 and 1.37 for winds from the SE and NE respectively. In further experiments, three sets of PG measurements were made above an even surface near sea level. Two of these experiments were at the Point of Trebister, 2 km SSE of the Observatory, and the other near the Sands of Sound, 1 km to the East (Figure 1c). In all, ten series of observations were obtained. The mean electrograph reduction factor computed from these was 1.36, very similar to the values obtained by the standard tests at the Observatory site.

## 3.2 1930-1939

In August 1930 a new type of collector was introduced, which consisted of polonium deposited on a copper rod, about 4 cm long by 0.5 cm diameter. These rods were recoated periodically through an arrangement with the Government Chemist, with a fresh collector fitted at the start of each quarter. Otherwise, PG records with the Benndorf electrograph continued as before, calibrated against absolute measurements made with the stretched wire and electrometer[11].

## 3.3 1940-1950

Benndorf electrograph measurements and standardisation with the stretched wire continued with little or no change of procedure[12]. Intermittent trouble was experienced with the electrograph, but recording continued with only a few breaks. The general behaviour of the Benndorf electrograph was improved in 1942 after replacing the sulphuric acid by glycerine, and the mica damping vane by a hook of copper wire. A new Wulf electrometer[13] was received in November 1948.

---

[11] The Wulf bifilar electrometers used for this were 5225 and 5716 (in 1931), and 5225 and 2965 (1932-39), manufactured by Günther & Tegetmeyer, Brauschweig, Germany (Fricke, 2011).
[12] (Hamilton, n.d.)noted a suggestion from Edinburgh in 1940 that the electrograph recordings should cease, but this did not occur as the superintendent, Oliver Ashford, felt that the length of the record was already sufficient for it to be worth continuing.
[13] Wulf electrometer No. 0157

204

### 3.4 1950-1970

205

Throughout this period, the Lerwick PG measurements began to show anomalous reductions, which also occurred at Eskdalemuir, Porto and Lisbon (Stewart, 1960). A reduction in PG due to radioactivity dispersal had been reported in Tucson, Arizona, following a nuclear detonation about 500 miles away in Nevada (Harris, 1955). It was concluded that the measurements at Lerwick were also affected by radioactivity contamination from distant nuclear weapons tests (Hamilton, 1965). When longer time series became available, a common effect was evident in data from the UK, Portugal and Japan (Pierce, 1972)[14], and radioactivity deposition processes were more extensively examined (Holzer, 1972).

During electrometer comparisons made in May 1960, variations in the reduction factors had been seen. This was investigated by a summer student in 1964, who found a marked PG variation with the wind direction. Greater PG values were found for southerly airstreams. By investigating air flowing over Loch Trebister and through measurements made during freezing conditions, the radioactivity effects were deduced to arise only over land. This conclusion followed from two observations. Firstly, during freezing conditions, which were assumed to prevent radioactive material leaving the soil, the PG was increased. Secondly, over water, which would not have retained surface radioactive material, the PG was also increased. Together, these indicated that the conductivity was only being reduced over land.

As part of the move to new offices in 1961, the Benndorf electrograph was dismantled. From January 1960, a thermionic valve electrometer designed by Dr Alan Brewer[15] replaced the Benndorf electrometer (Brewer, 1953) with a chart recorded added to provide the paper trace. This system is shown operating in Figure 6. A comparison was made between the Benndorf and Brewer electrographs from May 1960, and the values correlated well. It was concluded that, although the Benndorf was a simple instrument to use, its sensitivity was inconsistent across its range. In addition, the Brewer electrometer was more complicated, and it was thought to be more difficult to identify malfunctions. When repairs were needed, however, the electrometer valves were readily changed.

Up to October 1961, when readings were made at the stretched wire, values were subsequently taken from the Benndorf or Brewer electrometer charts. After October 1961, simultaneous readings were made at the stretched wire and the electrometer. The Wulf electrometer was also calibrated for each observation. Early in 1962 the sulphur insulators for the electrograph collector were replaced by PTFE, which were found to be entirely trouble-free.

---

[14] Similar effects were observed at Swider, Poland following the Chernobyl accident (Warzecha, 1987) and at Kakioka, Japan following the Fukushima accident (Takeda et al, 2011).
[15] Dr Alan W. Brewer (1915-2007), long-term collaborator of Prof G.M. Dobson at the University of Oxford, and instrument scientist.

There was also a broadening of interest to include air-earth current measurements, with a proposal to make continuous measurements at Kew, Eskdalemuir and Lerwick (Hamilton & Paren, 1967). Trials of air-earth current[16] apparatus began in 1969 (Dawson, 1978) using a well-insulated current-collecting plate, an electrometer current amplifier and a recording device. Incandescent light bulbs were used to provide some local heating to reduce the effects of moisture. Some analogue chart paper rolls from this period exist indicating a sustained period of evaluation, but the procedures were not sufficiently developed at that point to allow systematic tabulation.

## 3.5 1970-1985

The Brewer electrograph measurements continued in the established manner until July 1984. Experiments with conduction current density measurements continued, using instrumentation manufactured by Saxer and Sigrist. The sensor employed a collecting plate of area 0.5 m$^2$ mounted flush with the ground above a 30 cm deep slate-lined pit, situated between the met enclosure and the office block. The current from the plate was measured by the voltage developed across high value ($10^{10}$ $\Omega$ to $10^{12}$ $\Omega$) resistors, using a mechanical "Vibron" chopper semiconductor electrometer (see also Harrison & Nicoll, 2008). The final output voltage signal was passed to the Met Office standard "MODLE" (Met Office Data Logging Equipment) recording system. Tabulations of conduction current density were produced from July 1978.

## 4. Tabulations of data in the *Observatories Year Book*

The PG values were tabulated as monthly sets of daily values in the annual volumes of the *Observatories Year Book* until 1967, and thereafter on individual summary sheets taking a similar form with hourly values until 1984, stored in the National Meteorological Archive. The methods selecting representative values evolved during the twentieth century. Initially a geomagnetism-inspired approach was adopted, with a later method becoming established for selecting "fair weather" values in the second half of the twentieth century.

During the 1930s, the OYBs were published about two years in arrears, with the 1937 volume the last to be published before the war. The 1935 OYB contains photographs and a site plan, which show that, although no longer used, the original atmospheric electricity hut was still in position. Due to the war, the 1938 volume was published in 1955, and the 1939 volume, with the whole introductory section virtually omitted, in 1957. Because of the need to catch up from the war, only very brief introductions to the measurements were included in the OYB during 1950-1956. From 1957, a full introduction was included. The 1950-59 volumes were published in 1960 and 1961. Following reorganisation of geophysical observations in the UK, publication of the OYB by the Met Office ceased in 1967. The UK Atmospheric Electricity data were then published by USSR's Hydrometeorological Service, in their monthly issue of *Results of Ground Observations of Atmospheric Electricity*[17].

---

[16] The air-earth current in fair weather is also known as the vertical conduction current density, i.e. the vertical current flowing per unit area, and is typically ~ 2 pA m$^{-2}$.

[17] The final (1967) edition of the Observatories Year Book reported that this arrangement, organised through the World Meteorological Organisation, had begun in January 1964.

All atmospheric electricity measurements ceased at Lerwick Observatory in 1984, with the last monthly tabulation for July 1984.

## 4.1 Basic data record

The daily values included hourly mean values obtained over 60 min periods, centred at the exact hour GMT up until 1931, and on the half-hour thereafter. The equivalent PG in the open was provided, which was obtained by multiplying the chart reading by the reduction factor. Values were given for 03, 09, 15 and 21 GMT, and hourly on undisturbed days. When it was difficult to obtain a stable reading, typically due to precipitation, the entry was marked as "z", with a "+", "-", or "±" added to indicate the likely polarity of the mean value. In the tabulations, two sets of mean values were provided, that of

    (a)  All hours with positive values.
    (b)  The means for all days on which all the four six-hourly values were recorded.

Values during hours when the trace passed off the top of the chart were included in (a), the upper limit of registration being taken as the value for that period, i.e. essentially a saturation value. The range of the electrograph was about $\pm1500$ Vm$^{-1}$, only likely to be exceeded beneath strongly electrified clouds and thunderstorms.

## 4.2 Classification by "Electrical character"

Initially, the classification of the atmospheric electricity data was strongly influenced by the practices in magnetic recording, specifically that of assigning a description to the day's trace as quiet or disturbed. Following the same approach, the typical variations found in a day's recordings were classified by means of an "electric character figure", according to

        0 - a day (midnight to midnight) with no negative PG recorded,
        1 - a day with negative PG excursions totalling less than three hours,
        2 - a day with negative PG totalling more than three hours.

In 1927, "electric character letters" were added to the classification system. These were intended to show, in any of the hourly periods of the day:

        $a$ - that the PG range did not exceed 1000 Vm$^{-1}$
        $b$ - that the PG exceeded 1000 Vm$^{-1}$ at least once, but less often than six times
        $c$ - that the PG exceeded 1000 Vm$^{-1}$ for six or more times.

From 1927, the symbols ">" and "<" were introduced to designate that, during the measurement period, the PG had exceeded the range of the electrograph. When the measurement was estimated due to a defect, the value was enclosed in brackets.

From 1928, the electrical character description of each day was extended to include the duration of negative potential for each day. If the electrograph record failed but no precipitation had fallen, it was assumed that the PG had remained positive; if, however, precipitation fell when there was no record, no estimate was made

except when the missing segment was sufficiently small and any precipitation sufficiently continuous to allow reasonable interpolation.

In the OYB, a table of the greatest PG values (positive or negative) was given and a list of when the PG was negative for prolonged periods with only short excursions positive. From 1936 onwards, Lerwick sent, to Edinburgh for forwarding to Mr Gish[18] in Washington, annual tables giving the annual frequency of days of character 0, 1, 2, and monthly total duration of negative PG.

### 4.3 Classification by "Fair Weather"

In January 1957 the classification approach for the PG was fundamentally changed, to use the weather conditions at the time of the measurement. This offered an independent classification method for identifying times with minimal local effects on the measurements (Harrison & Nicoll, 2018). An hour was regarded as having "fair weather" (FW) conditions if four conditions were fulfilled:

> (1) there were no hydrometeors
> (2) there was no low stratus cloud
> (3) there was less than three-eighths cumuliform cloud
> (4) the mean hourly wind speed was less than 8 $ms^{-1}$.

Hours failing to meet the full FW criteria, but during which no hydrometeors (i.e. rain, snow, hail…) were observed, were also marked. A great advantage of this FW approach was, since these classifications were applied to each hour individually, daily mean values could still be determined even if the day was partially disturbed. Many more daily mean values could therefore be obtained despite disturbed conditions. From January 1957 to December 1966 when both classification systems were in use, 1879 daily values were found using the FW classification system, but for only 807 days by using the daily character figure method.

Continuous air-earth current measurements are available in the Met Office archive as tabulated data forms from July 1978 to July 1984. These are provided as hourly values in a similar manner to the PG, with values given for hours which were identified as "Fair Weather" or "No Hydrometeors".

Images of some Lerwick Observatory tabulated hourly record sheets from early and late in the measurements are given in Figure 7. The early record sheet (Figure 7a) was completed by hand, with values read directly from the electrogram trace; the later one is typewritten. In the later one (Figure 7b), the hours which did not meet the fair weather criteria were marked with a superscript "+". Some basic analysis, such as counting values and computing averages, is also included on the later record sheet.

---

[18] Oliver H. Gish (1900-1988), geophysicist working on atmospheric electricity and staff member of the Department of Terrestrial Magnetism at the Carnegie Institution of Washington. His work especially concerned the vertical column resistance of air, initially determined by the first stratospheric balloon flight, *Explorer II*. (Gish, 1944; Gish & Sherman, 1936)

## 5. Scientific findings from the Lerwick atmospheric electricity data

Little has been written about the overall scientific importance of the Lerwick atmospheric electricity data, but it is clearly an important atmospheric dataset which deserves full digitisation and further study. In the first decade of data, the PG measurements at Lerwick corroborated those obtained simultaneously by the *Carnegie* on its final voyage, such for 1st December 1928 (Figure 8a), and contributions continued to the Carnegie Institution of Washington's scientists into the late 1930s.

The most significant contribution during the time of the measurements was explaining that the appreciable PG reduction in the 1950s and 1960s, as illustrated in Figure 8b, was due to radioactive contamination following an intense period of northern hemispheres atmospheric nuclear weapons detonations. Lerwick's data was important for resolving this, as, unlike the companion site at Eskdalemuir, it was well removed from possible radioactivity releases from the nuclear site on the coast of Cumbria, then known as Windscale[19] (see also Figure 1a). It can only have been alarming for the scientists at Lerwick in the 1950s and early 1960s to observe major changes in a quantity which had shown consistent values for the previous three decades. Hence it is very much to their credit that the measurements were continued and investigated to find and report the cause (Hamilton, 1965). Although much less well known, it is not unreasonable to compare this analysis of surprising change with the discovery of the ozone hole over Antarctica in the 1980s. In both cases, data that were initially thought to be unsatisfactory for an unknown reason were explored for an explanation. Ultimately, extraordinary possibilities were justified by extraordinary evidence, and international agreements followed to address the environmental effects[20].

Since then, the thoroughness of the past atmospheric electricity and meteorological measurements made at Lerwick has allowed investigation of effects of enhanced radioactivity on the weather, leading to the finding of a small associated increase in rainfall (Harrison et al., 2020). A further discovery in the Lerwick PG data from the 1970s has been its strong link to climate, specifically to variations in Pacific Ocean temperatures associated with the El Niño–Southern Oscillation (Harrison et al., 2011). This illustrates the special value of data from a site where local influences are small, allowing global effects – in this case internal variability in the climate system – to be uncovered.

The decision to cease the atmospheric electricity measurements in 1984 almost certainly followed directly from the end of the measurements associated with the closure of Kew Observatory in 1981. In the announcement about Kew[21], mention is made of the loss of "a few specialist measurements (such as atmospheric electricity…and air pollution)". The 1984 cessation of Lerwick's measurements occurred when the value of long data series was relatively poorly appreciated in terms of environmental change, and came, unfortunately, shortly before the 1986 Chernobyl reactor accident for which the value of PG measurements would have been demonstrated again. Nevertheless, even with the truncated data set, it has been possible to derive new scientific

---

[19] This site was originally known as Windscale from 1956 – 81: it is now called Sellafield.
[20] The Partial Nuclear Test Ban treaty banning atmospheric nuclear tests was signed in 1963 (arguably hastened by the Cuban Mission Crisis of October 1962); the Montreal Protocol to phase out ozone-depleting substances was signed in 1989.
[21] *Meteorological Magazine* **109**, 215 (1980)

results, and Lerwick must be considered an exceptionally well characterised and suitable site for any future
atmospheric electricity monitoring.

**6. Conclusions**

Atmospheric electricity measurements in the UK have a long history, and were competently pursued at Kew,
Eskdalemuir and, especially, Lerwick over a long time. They are now of renewed relevance because of the
broader interest in climate-related quantities, which the Lerwick data have demonstrated includes the global
atmospheric electric circuit. For climate-related atmospheric electricity research, the PG measurements at
Lerwick are especially valuable because of the minimal interference from air pollution. The well-characterised
atmospheric electrical properties of Lerwick Observatory obtained over the majority of the twentieth century
strongly support the prospects of further measurements there, with the modern durable instrumentation now
available[22].

**Acknowledgements**

Figure 2 was provided by Alan Gair. Dr Hugo Silva (University of Porto) arranged access to the Benndorf
electrometer previously used at the Serra do Pilar Observatory, shown in Figure 4. Figure 5a and b are archive
material from Lerwick Observatory. Figure 7a and b were provided by the Met Office, and the Met Office
originally obtained the data shown in Figure 8. Further help was provided by Met Office staff: Norrie Lyall,
current Station Manager, Paul Nelson, Dr Graeme Marlton and Mark Beswick at the National Meteorological
Archive. Daniel Bennett (BBC Shetland) provided further historical sources.

**Data availability**

Scans of the annual volumes of the *Observatories Year Book* for Lerwick are available at
http://www.geomag.bgs.ac.uk/data_service/data/yearbooks/ler.htm

**Author Contributions**
RGH drafted the initial manuscript with help from JCR. Both RGH and JCR revised the manuscript.

**Competing interests**
None

---

[22] Importantly, modern all-weather field mills (e.g. Bennett & Harrison, 2007), and stretched wire measurements (e.g. Harrison, 1997), no longer have any need to use radioactivity.

**Figures and Tables**

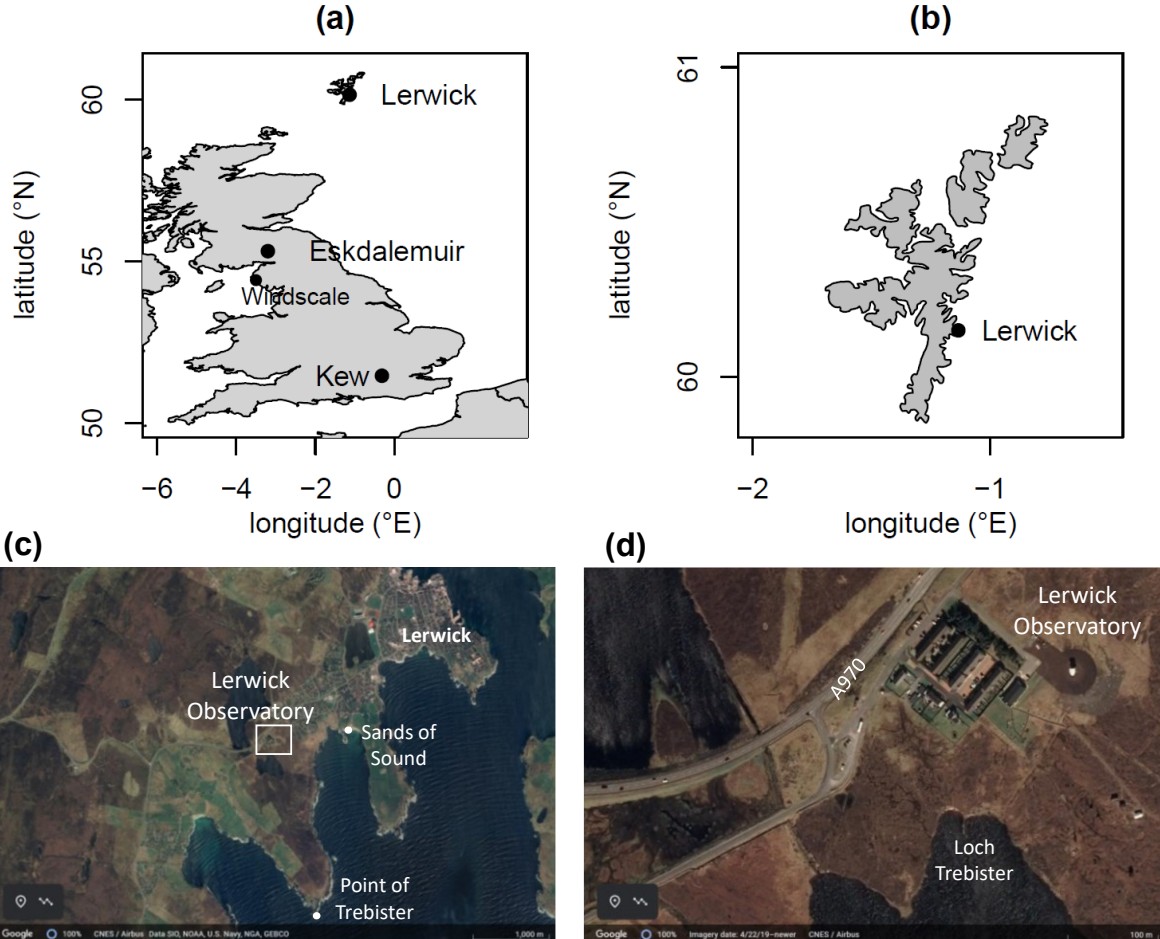

*Figure 1*. Maps showing: (a) the positions of the Geophysical Observatories at Kew, Eskdalemuir and Lerwick,
and the nuclear site previously known as Windscale (Sellafield), (b) Shetland, (c) and (d) the local environment
around Lerwick Observatory. *Images in (c) and (d) from Google Earth.*

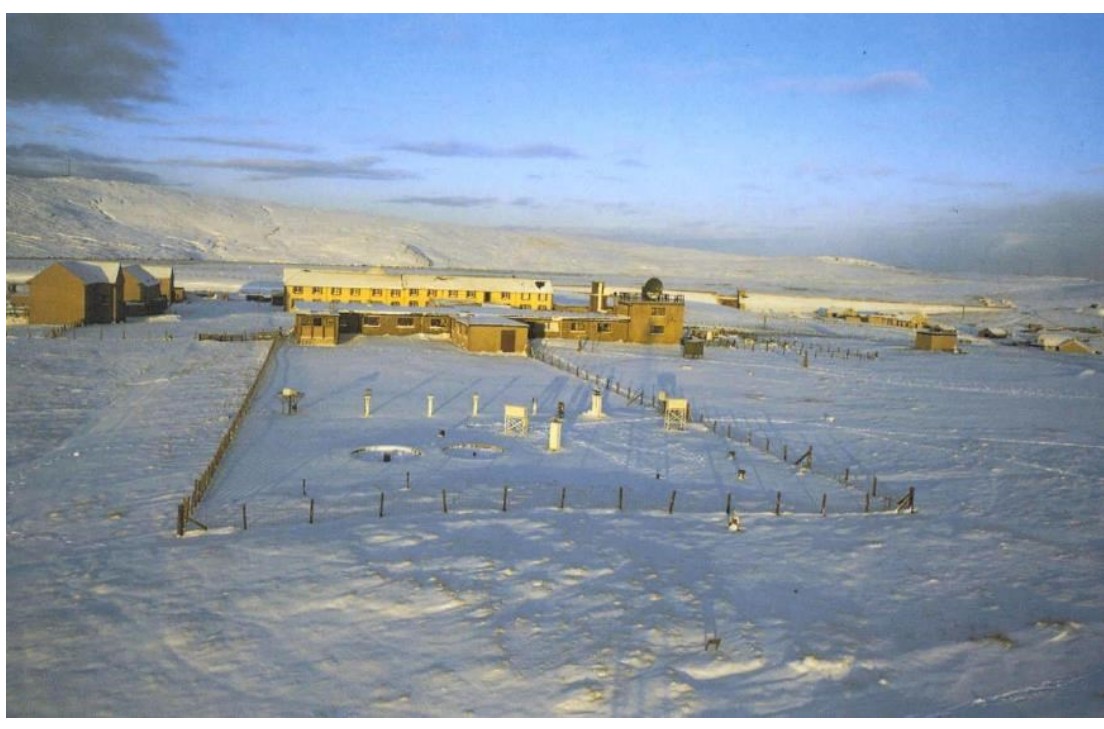

*Figure 2* The meteorological site and office buildings viewed from the south-east end, in the early 1990s.
*(Photograph provided by Alan Gair).*

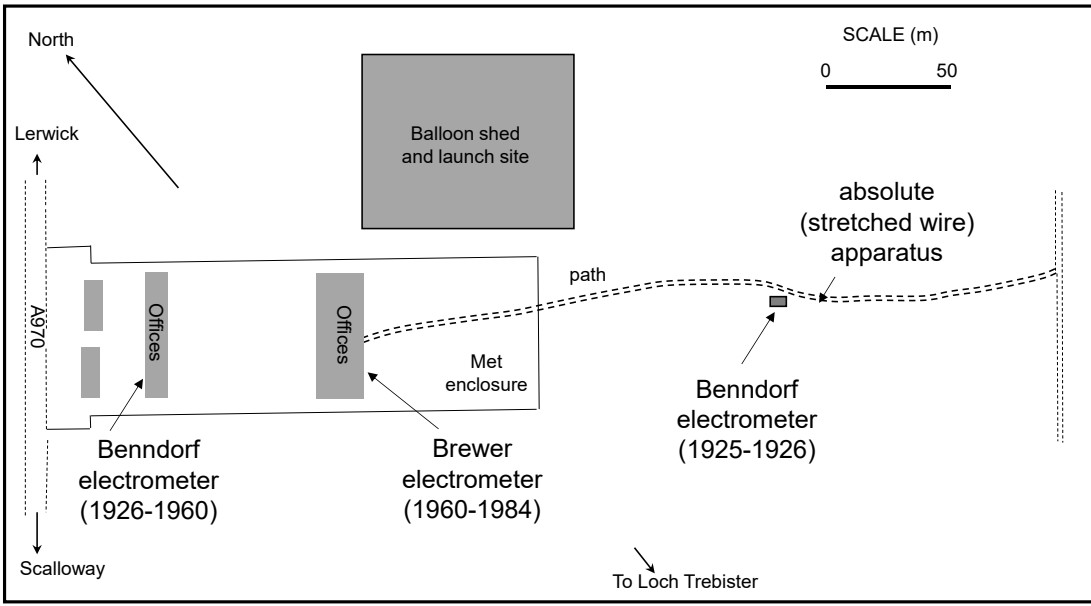

*Figure 3* Layout of the Lerwick Observatory site, from about 1961, based on OYB descriptions. The principal
positions of the atmospheric electricity sensors at different times through the measurement series are given.

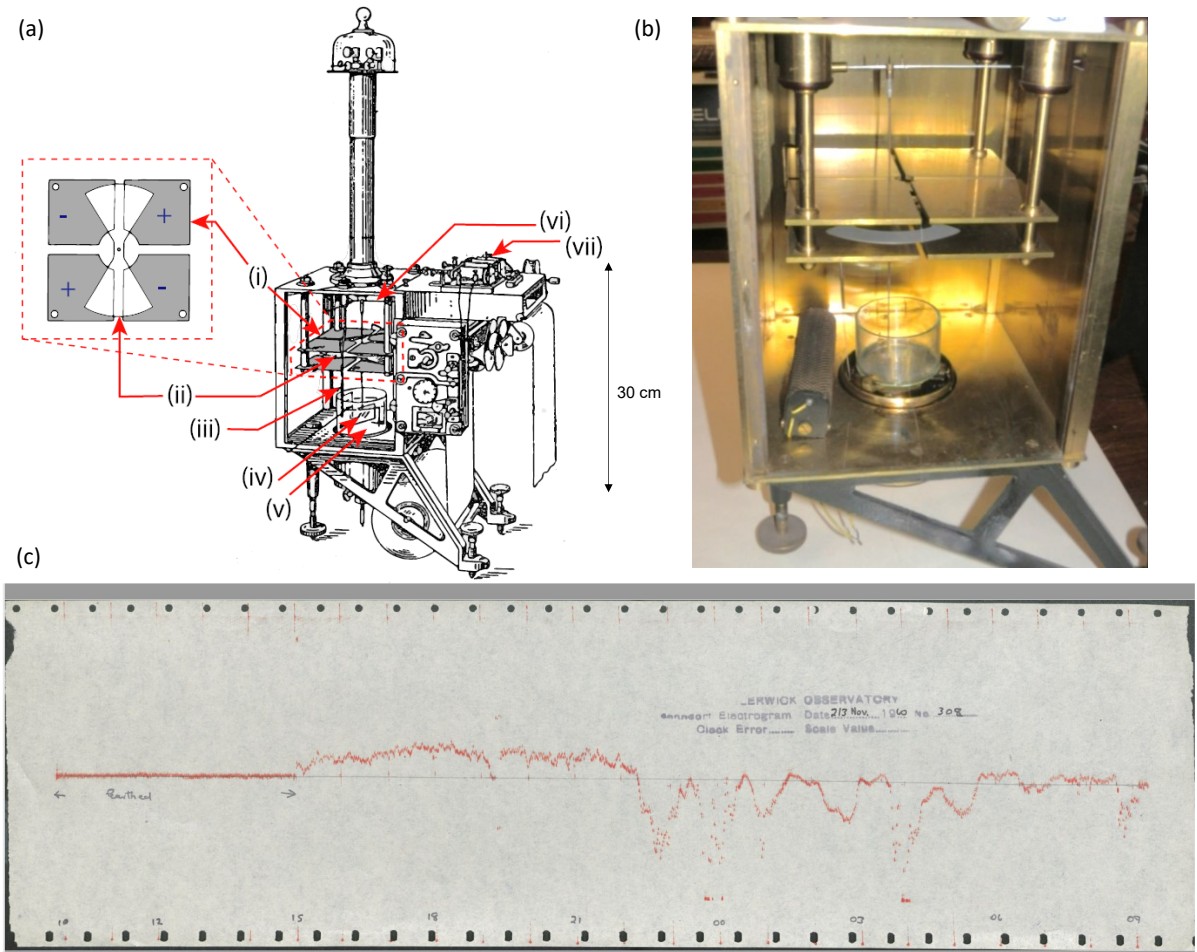

428 *Figure 4* The Benndorf Electrometer. (a) Schematic view of a Benndorf device with recording paper, modified
429 from (Nagamachi et al., 2022), showing the different components: (i) plate electrodes, (ii) quadrant plate, (iii)
430 connection wire, (iv) mica plate, (v) sulphuric acid pot, (vi) pen connecting rod, (vii) pen pressure adjuster. (b)
431 Internal view of the Benndorf electrometer used at the Serra do Pilar Observatory, Porto, showing the acid pot
432 and plate electrodes. (c) Benndorf electrometer paper chart, from Lerwick, for 2nd November 1960. *(Photograph*
433 *by Giles Harrison, and Benndorf trace from the National Meteorological Archive).*

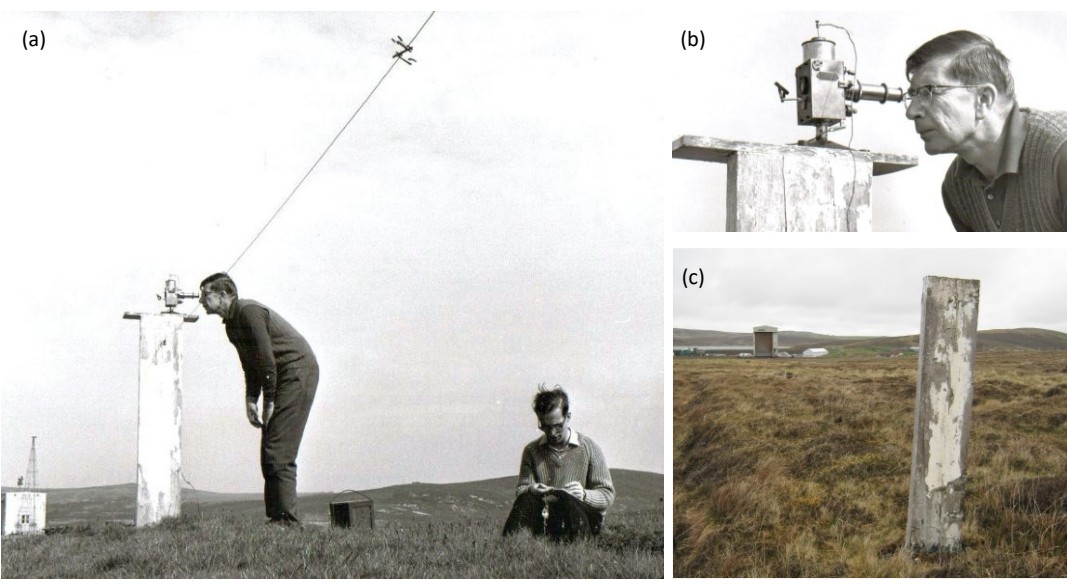

*Figure 5*. (a) Use of a stretched wire, insulated at each end, to determine the potential at 1m in the open (*c*1963) with an electrometer. The Lerwick radiosonde balloon shed is in the background and a radioactive source has been clipped to the midpoint of the wire. Observatory superintendent Richard Hamilton is operating the electrometer, with a student assistant. (b) Detail of the electrometer and crocodile clip connections. (c) One of the stretched wire posts still survives intact, with the remains of the electrometer support post in (a) also evident (image from May 2022). *Photographs in (a) and (b) from Lerwick Observatory archive, (c) provided by Norrie Lyall, Lerwick Observatory.*

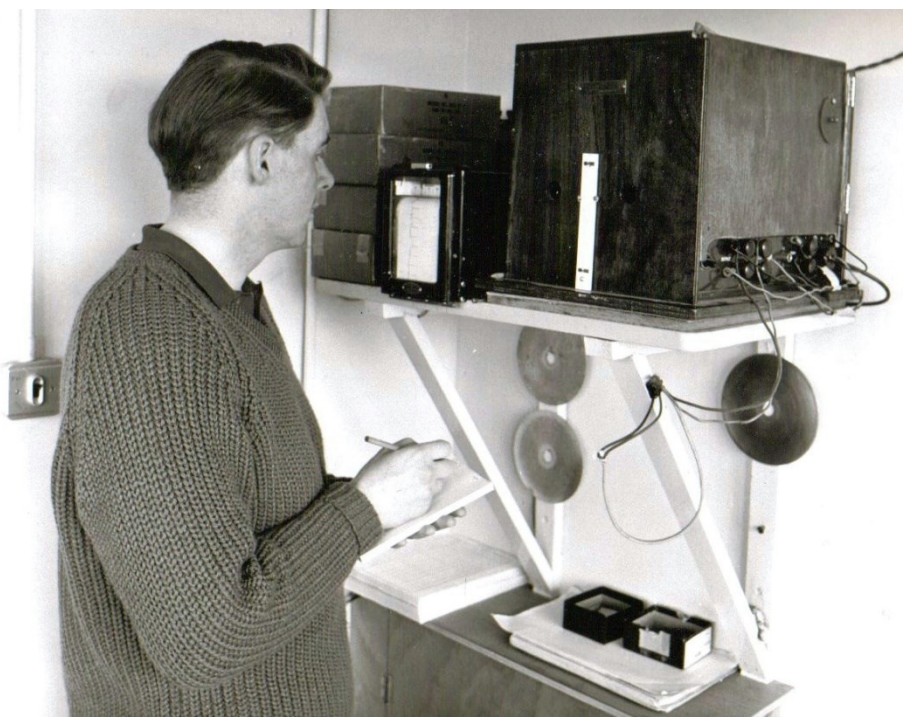

*Figure 6*. Readings being taken from the Brewer valve electrometer's chart recorder, by Monty Georgeson, a scientific assistant at Lerwick Observatory from 1966. *Photograph from Lerwick Observatory archive.*

Met O 1, Form 31
(May 1971 Edition)

POTENTIAL GRADIENT (close to the ground, over an open level surface).
Mean values for hours without hydrometeors and for fair weather hours.

Observatory Lerwick     Factor 1·85     Month August     Year 1982

| Hour GMT | 0–1 | 1–2 | 2–3 | 3–4 | 4–5 | 5–6 | 6–7 | 7–8 | 8–9 | 9–10 | 10–11 | 11–12 | 12–13 | 13–14 | 14–15 | 15–16 | 16–17 | 17–18 | 18–19 | 19–20 | 20–21 | 21–22 | 22–23 | 23–24 | Mean |
|---|---|---|---|---|---|---|---|---|---|---|---|---|---|---|---|---|---|---|---|---|---|---|---|---|---|
| | | | | | | | | | | | | volts per metre | | | | | | | | | | | | | |
| 1 | | | | | | | | | | | +10 | +30 | +20 | | | | | +165 | +140 | +150 | +175 | +150 | +100 | | |
| 2 | +75 | +65 | 65 | 75 | +75 | +100 | +100 | +95 | +20 | +10 | +10 | +10 | +0 | +0 | +20 | +35 | +45 | +55 | +75 | +85 | +65 | | 55 | 75 | |
| 3 | 95 | 100 | | | 110 | 155 | 130 | 120 | 140 | 155 | 175 | 215 | 140 | – | – | 195 | 240 | 215 | +240 | | | | | | |
| 4 | | | | | +380 | +340 | +285 | 230 | 215 | 155 | 130 | 140 | 130 | 140 | 130 | 130 | 110 | 130 | 140 | +175 | +215 | +205 | +285 | | |
| 5 | | | | | | | | +220 | 195 | 150 | 130 | 205 | 100 | 100 | 100 | 95 | 110 | 110 | 100 | 140 | 120 | +100 | | | |
| 6 | | | | | | | | | | | +140 | +120 | | | | +140 | +140 | +120 | +110 | +120 | +130 | +110 | +120 | | |
| 7 | +100 | +100 | +95 | +95 | +95 | +110 | +95 | +100 | +85 | +130 | | +120 | +130 | +95 | +95 | +130 | +65 | +120 | +155 | +230 | +55 | | | |
| 8 | | | +45 | | | | | | | | | | | | | +55 | –95 | –95 | –120 | –140 | | | | |
| 9 | –85 | –30 | –20 | 0 | | | +0 | +0 | +–35 | +–35 | +–45 | +–30 | +–45 | +–55 | +20 | | +– | | | | | | |
| 10 | | +–20 | | | | | | | | +–45 | | | | | | | +–85 | | +–85 | +–45 | +–45 | +–30 | +–35 | +–10 | |
| 11 | | | | | | +100 | +75 | 75 | 100 | +75 | +95 | | | | | +140 | +305 | +– | +– | +– | +370 | +100 | | |
| 12 | +75 | +95 | +95 | +95 | +95 | | +0 | +0 | +0 | +0 | | | +–10 | | | | | | | | | | | |
| 13 | | | +65 | | | | | | | +85 | | +100 | +100 | +110 | | | | | | 100 | | | | |
| 14 | 100 | – | | | | – | 110 | 120 | 110 | 95 | 100 | 95 | 85 | 95 | 95 | 100 | 95 | 95 | | | | | – | |
| 15 | – | – | 120 | +155 | +130 | +195 | +285 | +250 | +240 | +425 | +405 | +325 | +260 | +215 | +205 | +175 | +185 | | | | | | | |
| 16 | | | | +380 | +400 | +425 | +230 | | | +165 | 185 | 215 | 335 | 285 | 240 | | | +140 | +120 | 120 | | | |
| 17 | 120 | 110 | 110 | 140 | 140 | | | | +165 | +140 | +150 | 130 | +140 | 150 | 185 | 260 | 240 | +240 | 185 | 205 | 335 | 270 | | 250 | |
| 18 | | +370 | +415 | +490 | +435 | | | | | | | | | | | | | | | | | | | |
| 19 | | | | | | | | | | | | | | | | | | | | | 155 | +130 | | |
| 20 | 75 | | | 65 | | | | | | | | | | | 140 | 120 | | | | | | | | |
| 21 | | | | | | +10 | | | | | | | | | | +10 | 10 | 10 | 10 | 10 | 10 | 20 | 10 | 30 | |
| 22 | · | | 10 | 10 | 10 | 35 | 20 | 10 | 0 | | | | | | | | | | | | | | | | |
| 23 | | 20 | 10 | 10 | 10 | 35 | 100 | 75 | 100 | 140 | 100 | 120 | 120 | 130 | 120 | 140 | 140 | 150 | 130 | 75 | 45 | | | |
| 24 | | | 10 | 65 | 55 | 75 | +130 | +120 | +110 | +95 | +120 | +100 | +110 | | | | | | | | | | | |
| 25 | 45 | 35 | 35 | | | | | | | | | | | +65 | | | | | | | | | | |
| 26 | | | | +165 | | | | | +110 | +65 | +75 | +55 | +30 | | +165 | | | +120 | +85 | +100 | | | | |
| 27 | | | | | | | | | 100 | 100 | 110 | 150 | 110 | 110 | 100 | 95 | 95 | 100 | 150 | 140 | | | | |
| 28 | +95 | +95 | +95 | 85 | 75 | | | | | | | | | | | | | | | | | | | |
| 29 | 150 | 175 | 230 | | | | | | +220 | +245 | +195 | +185 | +240 | +305 | +445 | +445 | | | | | | | | |
| 30 | | | | +335 | | +490 | | +30 | +35 | | | | | | +30 | +20 | | | +20 | | | | | |
| 31 | | | | | | | | | 85 | 120 | 95 | 130 | 120 | | | | | | | | | | | |
| Mean | (11) 75 | (12) 95 | (13) 100 | (13) 105 | (13) 135 | (10) 180 | (12) 140 | (13) 150 | (14) 105 | (13) 120 | (16) 100 | (17) 105 | (18) 110 | (16) 105 | 100 | (15) 130 | (17) 145 | (21) 125 | (19) 130 | (13) 115 | (12) 105 | (14) 110 | (16) 130 | (15) 95 | (349) 115 |
| Fair Weather Mean | (7) 70 | (6) 70 | (8) 70 | (8) 55 | (6) 65 | (4) 70 | (4) 115 | (4) 135 | (5) 135 | (6) 130 | (6) 135 | (7) 130 | (6) 115 | (7) 130 | (8) 145 | (10) 145 | (11) 145 | (9) 105 | (9) 80 | (6) 105 | (6) 95 | (7) 100 | (7) 110 | (7) 105 | (164) 105 |

The potential gradient is reckoned as positive when the potential increases upwards. The small + denotes a non-fair weather hour. No entry is made for hours with hydrometeors and dashes are inserted for hours of defective record. The number of hours or days used in computing each mean is shown in round brackets.

Met.O/Carto/D.O.J 1342

Figure 7 Observatory record sheets for (a) December 1947 and (b) August 1982. In (a), the overall character of a
449 day's measurements is given at towards the top of each daily column, and, in (b), an hourly classification was
450 applied for the presence or absence of fair weather conditions.

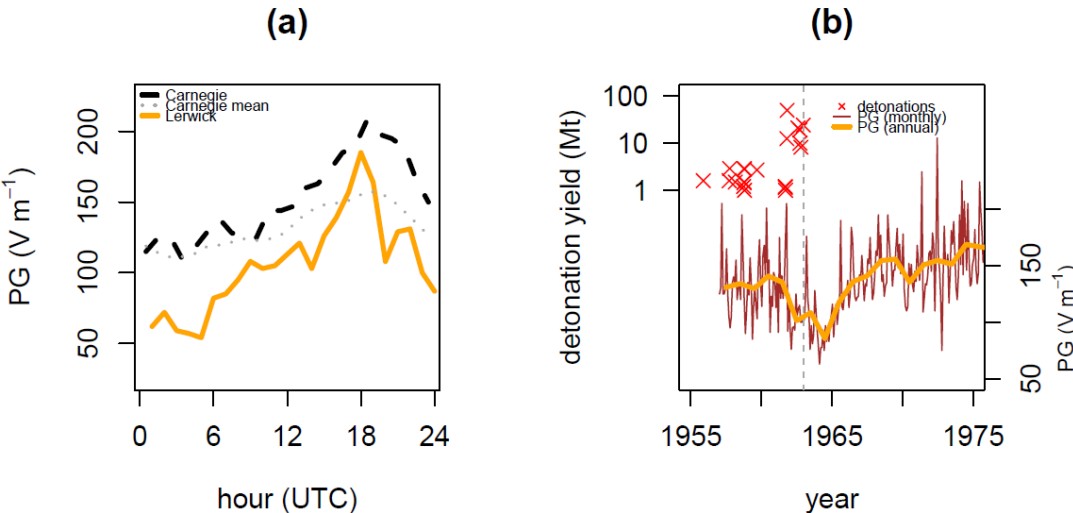

*Figure 8* (a) Hourly values of PG on 1st December 1928, on board the *Carnegie* which was then at 29°S, 245.2°E
(solid line), and recorded simultaneously at Lerwick (dashed line). Thin dotted line shows the averaged values
from the *Carnegie* for Cruise 7, (data from Harrison, 2013). (b) Northern hemisphere nuclear weapons
atmospheric detonations greater than 0.5 Mt (crosses, from Warner and Kirchman, 2000), and Lerwick fair
weather PG as annual (thick line) and monthly (thin line) means. The vertical dashed line marks the beginning
of 1963: the Partial Nuclear Test Ban Treaty was agreed in August 1963.

Table 1 Summary of events in operation of the Lerwick atmospheric electricity apparatus

| date | Aspect | source[23] |
|---|---|---|
| 1921 June 7th | Lerwick Observatory opened | OYB28 p24 |
| 1926 | • Benndorf electrograph installed (no 108).<br>• Initially housed in a small wooden hut, oil stove heated. Collector rod passed through NE corner of hut. (p23)<br>• Scale factor 0.78, determined against stretched wire with burning fuse and electroscope or electrometer. (p24).<br>• Instrument is "sluggish" in comparison with a Kelvin water dropper, especially when calm, although such circumstances unusual at the Lerwick site (p24).<br>• Summary data available Jan to July (p25) | OYB26; OYB27 p24 |
| 1926<br>6th to 24th July | • Hut inconveniently small and generated oil fumes so Benndorf electrograph moved to NW corner of office block. (Scale factor of new site 1.53).<br>• New position 236 m from position of absolute determinations of reduction factor | OYB26 p23 and p24; OYB27 p24 |
| 1926 24th July | Benndorf instrument sent for overhaul (from 24th July to Dec) | OYB26 p23 |
| 1927 | Electrograph collector passed through a window in north wall, 1.9m from corner of building. Collector is 4.76m above the ground and protrudes 1.23m from window. Consists of a copper spiral 5cm long coated with radium salt. | OYB27 p24 |
| 1928 | • Mean reduction factor for the year 1.31, 3% lower than for the mean of the 1927 factors, also found to vary slightly with wind direction.<br>• Rise time to half scale deflection is 69 s (from 140 tests during 1927); 10x slower than Eskdalemuir water dropper<br>• Leak tests show half the potential lost in 38.5 mins. (Insulation is worse during Aug to Oct due to insects; excluding these gives a half time of 50 mins). | OYB28 p27 |
| 1928 June 26th | Collector replaced after gale damage | OYB28 p26 |
| 1928 Dec 7th | More rapid collector installed. Time to half deflection 48 s (from 13 tests) | OYB28 p26 |
| 1929 | Tests on collector – time to half deflection 45 s | OYB29 p24 |
| 1930 June | Tests on collector – time to half deflection 33 s | OYB30 p24 |
| 1930 Aug 16th | Collector changed from the previous spiral of wire carrying radium sulphate, to a copper rod (5 cm by 0.5 cm, tapped at 2 BA) with polonium plated on the unthreaded end for 12 mm. (These were to be recoated periodically by the Government Chemist, with a fresh one fitted at the beginning of each quarter). | OYB30 p23 |
| 1930 September | Tests on new collector – time to half-scale deflection 4 s | OYB30 p25 |
| 1941 Dec | Insulation had been poor for several days. Amber insulation cleaned with alcohol (10th Dec). Still bad, so sanded and painted with ether (12th Dec). Insulation good from then on. | Handwritten notebook from Lerwick, unnamed |
| 1942 Jan 2nd | Collector changed "after seal test" | |
| 1942 May 13th | Corrosion evident in instrument. Resistance coils open circuit, sent for repair; quadrant electrometer working throughout. | |
| 1942 May 16th | Acid spilled inside electrometer. Rubber amber lightly after sandpaper all metal parts. "Leak normal after reassembly." | |
| 1942 May 26th | Leak high. Amber supporting acid pot treated with alcohol. | |
| 1942 June 4th | Battery renewed 1550 | |
| 1942 June 20th | Seals test 0934. Ribbon reversed. Restarted 1015. | |
| 1942 Aug 13th | Platform supporting acid pot again loose. Dismantled. Platform secured then soldered to the screw heads already attached to the | |

---

[23] OYB*yy* refers to the annual volume of the *Observatories Year Book* for the year 19*yy*.

| | amber. | |
|---|---|---|
| 1942 Aug 16th-22nd | Leak high every day. Amber cleaned and sandpaper until satisfactory. | |
| 1942 Sep 2nd | Leak high. Amber cleaned. | |
| 1942 Sep 4th | Leak high. Amber cleaned. | |
| 1953 January | Collector blown away in a blizzard. New one dispatched within 24 hours. Lost one ultimately recovered, bent but serviceable. | Nature 4466, 965 (1955). |
| 1959 | Tests on new collectors show that they have 80 to 200 µCi<br>Fresh collectors have a half time of 4 to 6 s. This decreased with operating time, probably due to weather damage. Regular replacement leads to half-times not exceeding 20 s. | OYB57 p14<br><br>OYB57 p15 |
| 1959 22nd Oct to 24th Oct | Ion production rate from b ionisation measurements made at 5cm over wet grass. Found to be about 10 ion pairs $cm^{-3}$ $s^{-1}$ | (Stewart, 1960) |
| 1961 Jan 1st | Brewer thermionic valve electrograph (Brewer, 1953) (which had been running in parallel), fully replaced the Benndorf electrograph. | OYB61 p13 |
| 1961 July 13th | Brewer electrograph moved into new building. Boom projects 58 cm from NE wall of electrograph room, at 2.06 m above the ground, and 160 m from the absolute PG measurements site. | OYB61 p13 |
| 1962 August 31st | Collector boom and recorder moved from a temporary position in anemometer room | (Hamilton, n.d.) |
| 1963 January | Collector boom and recorder installed in the Ozone extension. | |
| 1969 | Trials begin of air-earth current apparatus. | (Dawson, 1978) |
| 1978 July | Monthly tabulated values of air-earth current density begin | (Harrison & Nicoll, 2008) |
| 1979 Jan to 1980 Jan | Overlapping period of air-earth current measurements at Kew and Lerwick; median values in fair weather 1.2 pA $m^{-2}$ and 2.5 pA $m^{-2}$ respectively | (Harrison & Nicoll, 2008) |
| 1984 July | Last month of tabulated data of PG and air-earth current density | (Harrison & Nicoll, 2008) |

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
