# Peer review of "Atmospheric electricity observations at Lerwick Geophysical Observatory"

_History of Geo- and Space Sciences, 2022_

## Author Response (AR1)

**Responses to reviewer comments**

*hgss-2022-6: Atmospheric electricity at Lerwick Geophysical Observatory (Harrison and Riddick)*

We are grateful to the editor and reviewers for the attention to the manuscript and the comments made. We have revised the manuscript accordingly, with the principal changes marked on the manuscript in red.

**Response to Editor's comments:**

*This is a very interesting and valuable manuscript. I have just two suggestions:*

1. *It would be instructive to include a few examples of the instrumentation recordings (scan of paper recordings) of the measured electrical quantities, if available.*

We agree and we have been able to locate a Benndorf electrometer chart, which has been added to figure 4. For completeness, we have also added figure 7, which shows Observatory record sheets from near the beginning and end of the records.

2. *When mentioning the Wulf electrometer its manufacturer "Günther & Tegetmeyer, Brauschweig, Germany" should be added.*

Thank you – this has been added to footnote 11.

**Response to Reviewer 1's comments:**

*This paper presents the history of the atmospheric electricity measurements at Lerwick Geophysical Observatory. In particular, there is a detailed description of efforts to record the atmospheric potential gradient at the Observatory and maintain high quality of the measurements. From this point of view, the paper is comprehensive, and contains many interesting details.*

Thank you.

*Perhaps examples of recordings could be also shown in other figures.*

We have now added a Benndorf chart record to figure 4, and given examples of the record sheets used in figure 7.

*Less has been said on contemporary atmospheric measurements throughout Europe and worldwide, albeit this is not the main subject of the article. Some background, in addition to the situation with the atmospheric electricity measurements in the UK, could have been added in the introduction.*

An introductory sentence giving some more context has been added, together with re-ordering of material.

*There is only a couple of technical errors, which can be easily dealt with on the final preparation of the manuscript for publication, e.g. line 123: "Figure 4" could be put in brackets, line 200: a space missing after "tests". In addition, in line 237, at least one of the range values of the resistor's resistance should be corrected.*

Thank you – changes made.

**Response to Reviewer 2's comments:**

*This is a well-organized, clear and lucid work, that describes the history of electrical measurements conducted at the Lerwick observatory in the Shetland Islands. The text is supported by images and drawings and gives a splendid description of the achievements of that remote station.*

*I still find some issues that require elucidation, and I would suggest a minor revision is made to the present text. Some clarifications are also needed as detailed below. After these are responded to, the manuscript can be accepted for publication.*

*Major Comments*

*The title of the manuscript does not include the word "observations" (or "measurements"). Clearly there is "atmospheric electricity" in Lerwick and anywhere else, but what makes this place unique is the fact that actual scientific observations were made there for a prolonged period. I suggest that the authors consider adjusting the title of the manuscript accordingly.*

We agree, and we have accordingly added the word "observations" to the title.

*At least some quantitate data and perhaps an original record from Lerwick should be presented. What was the average (fair weather "background") PG value measured there and was there indeed a clear diurnal Carnegie-like cycle? This should come after section 4.3.*

We have sought to do this through modifying figure 4, and adding figures 7 and 8. Figure 7 shows some data tabulations and figure 8a shows an unusual example of the Carnegie curve on a single day, as, fortuitously, both the *Carnegie* and Lerwick Observatory experienced undisturbed weather on 1st December 1928.

*I salute the author's bold parallel between the detection, at Lerwick, of nuclear tests' effects on atmospheric electricity and the discovery of the ozone hole in the 1980s by the British Antarctic Survey at Halley Bay in Antarctica (lines 339-340). I whole-heartedly accept this analogy and support the authors' quest for renewing electrical observations there.*

We appreciate the reviewer's support for the parallel drawn between diagnosing the atmospheric effects from nuclear tests and the ozone hole, and we have included a sentence summarising this aspect in the Abstract.

*However, this part requires elaboration:*

*Concerning the effects of radioactive contamination on the PG measurements, the readers would benefit from graphs showing the onset of the electrical changes observed at Lerwick, perhaps marking specific dates of nuclear explosions. See for example Huzita (1966) that showed data obtained after the large soviet nuclear tests at Novaya Zemlaya on October 30, 1961. Pierce (JGR 1972) reported changes in PG from 6 stations, including the UK, for the period 1950-1964.*

We are limited in the data analysis we can provide, given the intended historical scope of this article, and the fact that the majority of the data has not yet been keyed, but we have been able to add some related material. Fig 8b shows a time series of the larger northern hemisphere air detonations, and the response of the fair weather PG at Lerwick around the same time.

*Lines 217-223 - It is confusing that the unnamed "summer student" reported a 6% increase to the PG, while the opposite should be expected if the conductivity is increased due to a radioactive source being introduced to the area (as per Ohm's law). Reference should be made to the work of Lee Harris (JGR, 1955) and Holzer (JGR, 1972).*

This section has been reworded; the intention is to describe how the local environmental aspects were used to help explain the effects observed. Both suggested references are now cited. We also mention, in a footnote, that similar effects were found at other sites after the Chernobyl and Fukushima reactor accidents.

*The statement in line 219 "This was probably linked to the surface radioactive deposition" in 1964 and "some confirmation was found…the freezing (of the lake) in February 1966" and the short mention of the direction of the prevailing winds there seem a little speculative. As this paragraph holds key importance to the topic above, a better and more in-depth explanation is needed.*

This material has been rewritten to improve clarity, and slightly reordered. The importance of investigating the Loch and freezing conditions was that it revealed the origin of the radioactivity effects. In the case of frozen land, which is expected to prevent radioactive material leaving the soil, the PG was increased. Over water, where there would have been no deposition retained at the surface, the PG was also increased. Together, these observations indicated that the conductivity was only being reduced over land.

*Minor comments*

*In section 2.2 I recommend adding a map showing the location of Shetland and specifically the location of the Lerwick observatory, to give readers a better geographical context. This is highly needed as the text refers, for example, to Loch Trebister, Point of Trebister and to the Sands of Sound, when describing the different components of the observatory (lines 68, 179, 180). Also the location of the Windscale test site (line 336) should be marked.*

A multi-section set of maps has been added, as figure 1. This shows the other UK sites, including Windscale. Google Earth satellite images are used to show the detailed region around the Observatory itself to allow the specific locations mentioned to be identified. We have also clarified that Windscale has since been renamed to Sellafield.

*The description of the various houses (buildings) in lines 64-70 and their numbers is not matched against either Figure 1 or 2, which do not overlap and make orientation quite difficult. A better description is needed to coordinate between the text and the image and drawing.*

The houses were ultimately demolished, so cannot be made to correspond directly to the later diagram. As it is only the typical uses of the buildings which was being addressed, the house-by-house details in the text are unnecessary and have been removed. The linking material to figure 3 has been improved.

*Line 255 – was the Soviet Union (USSR) indeed responsible for publishing the UK atmospheric electricity data? One wonders how this came about. Please elucidate this intriguing fact.*

Some more text has been added on this point. International atmospheric electricity data was published by the USSR's Hydrometeorological Service through an arrangement made by the World Meteorological Organisation.

*Section 4.2, lines 285-287 – it is interesting to learn if there were instances in the data when the PG exceeded the threshold of 1 kV/m? One wonders what would cause such strong deviations from average values and whether they are related to fog, volcanic ash or pollution.*

This is something to be considered further when the full dataset is available digitally. The largest deviations from the electrograph's measurement range would almost certainly be associated with strongly electrified clouds overhead, thunderstorms and/or charged rain reaching the surface. This has been mentioned at the end of section 4.1.

---

## Author Response (AR2)

**Further author responses**

*hgss-2022-6: Atmospheric electricity at Lerwick Geophysical Observatory (Harrison and Riddick)*

Thank you for the further consideration of our revised manuscript. In response to the points made:

- Figure 7. This shows images from archive material of the National Meteorological Archive. An acknowledgement following the form they have asked for, following an enquiry, has now been included in the caption.
- Figure 1. Google Earth material added to caption as requested.
- Author names. These are now given in full. Note, please, for the first author, that his first name is not used, hence just the first initial is conventionally written.